# The Impact of Irrigation on Olive Fruit Yield and Oil Quality in a Humid Climate

Paula Conde-Innamorato [1,*], Claudio García [1], Juan José Villamil [1], Facundo Ibáñez [1], Roberto Zoppolo [1], Mercedes Arias-Sibillotte [2], Inés Ponce De León [3], Omar Borsani [4] and Georgina Paula García-Inza [1]

1   Estación Experimental INIA Las Brujas, Programa Nacional de Investigación en Producción Frutícola, Instituto Nacional de Investigación Agropecuaria (INIA), Canelones 90200, Uruguay; cgarcia@inia.org.uy (C.G.); jjvillamil@inia.org.uy (J.J.V.); fibanez@inia.org.uy (F.I.); rzoppolo@inia.org.uy (R.Z.); ggarciainza@inia.org.uy (G.P.G.-I.)
2   Unidad de Ecofisiología de Frutales, Departamento de Producción Vegetal, Facultad de Agronomía, Universidad de la República (UDELAR), Garzón 780, Montevideo 12900, Uruguay; marias@fagro.edu.uy
3   Departamento de Biología Molecular, Instituto de Investigaciones Biológicas Clemente Estable, Montevideo 11600, Uruguay; iponce@iibce.edu.uy
4   Laboratorio de Bioquímica, Departamento de Biología Vegetal, Facultad de Agronomía, Universidad de la República, Montevideo 12900, Uruguay; oborsani@fagro.edu.uy
*   Correspondence: pconde@inia.org.uy; Tel.: +598-99120423

**Abstract:** The expansion of olive orchards into regions with no tradition of olive production and humid climates, such as Uruguay, with more than 1200 mm of annual rainfall, calls into question the need for irrigation. In these regions, however, years with water deficit during summers are quite common. The vapor pressure deficit during summer is lower than in countries with a Mediterranean climate. The high variability in interannual water availability in the current context of climate change, with a growing tendency for extreme events to occur, emphasizes the need to evaluate the production response of olive trees to irrigation. To achieve this, three irrigation treatments were applied to Arbequina and Frantoio cultivars according to the value of the maximum crop evapotranspiration: a first treatment applying 100% ETc, corresponding to being fully irrigated; a second treatment applying 50% ETc; and a third treatment in which neither irrigation nor rain inputs occurred from the end of the pit hardening period until harvest. Results show the possibility of an increasing fruit weight and pulp/pit ratio through irrigation in the local environmental conditions. The oil content in response to irrigation was different within cultivars. Water restriction conditions did not affect the oil content of olives in Arbequina, while in Frantoio it increased it. Polyphenols in fruit increased under water stress for both cultivars. The technological applicability of the results obtained must be accompanied by an economic analysis. The results obtained highlight the need for better use of irrigation water during the growth and ripening phase of the olive fruit under a humid climate.

**Keywords:** *Olea europaea* L.; drought stress; stem water potential; fruit growth; oil content; polyphenols

## 1. Introduction

The expansion of olive trees into new climate areas where temperature and precipitation regimes are different from those of the Mediterranean basin generates uncertainty regarding their ecophysiological response and represents challenges in crop management [1]. In temperate humid regions such as Uruguay, where annual precipitation is around 1200 mm, the need for irrigation is questioned. However, Uruguay presents high interannual climate variability and an irregular rainfall distribution throughout the year, which generates periods of water deficit [2,3]. In addition, these extreme events are expected to be more frequent [4], affecting productive behavior.

The importance of local evaluations has been highlighted by several authors who place emphasis on vapor pressure deficit (VPD) conditions [1,5,6], which is lower in Uruguay

than in the Mediterranean region. Lower VPD values are associated with lower tree transpiration and, consequently, lower water consumption. In this context of variability of water supply and environmental characteristics, it is necessary to evaluate the productive response of olive trees under different water status conditions, identifying the best balance between yield, oil quality and water productivity.

Olive has a high capacity to grow under water scarcity conditions due to its morphological characteristics and physiological mechanisms, related with the escape, avoidance and tolerance components of stress resistance [7,8]. However, there is a lot of studies that confirm that this crop positively responds to irrigation. Irrigation increases vegetative shoot growth, as well as final fruit size and yield [9–15].

Olive fruit growth (expressed as fresh weight) follows a double sigmoid curve [16]. Previous reports have identified two periods during fruit growth that are particularly sensitive to water restriction: an initial one during cell division and the expansion phase, from flowering until the end of fruit set, where deficit irrigation can reduce the final fruit number [15,17]; and a second one, during cell expansion and the lipogenesis phase, after pit hardening until harvest, when fruit growth increases sharply as mesocarp cells expand. A deficit in irrigation during this period can reduce the final fruit weight and oil content, and it can affect the oil composition, such as the polyphenol content [18–20].

There are several studies on the effect of water restriction on olive trees in arid regions, but there is a knowledge gap on the response of olive trees to irrigation management in humid temperate climates. The aim of this work was to quantify the impact of different irrigation regimes on fruit growth development and oil quality in two olive cultivars in a humid climate region.

## 2. Materials and Methods

### 2.1. Plant Material and Experimental Design

The experiment was conducted at the INIA "Las Brujas" Experimental Station in southern Uruguay (34°40′ S; 56°20′ W; 32 m above mean sea level) using the Arbequina and Frantoio cultivars. The olive trees were planted in 2006 at a density of 416 trees per hectare (4 m between trees and 6 m between rows) and were trained as single-trunk vase shapes, with three to four main branches. The orchard was managed as a commercial farm. Pest management was performed according to the Integrated Pest Management guidelines [21]. For each cultivar, a randomized complete design with three irrigation treatments and four replicates was used. The experimental unit is the tree and there are four trees per treatment, for each cultivar. Three irrigation treatments were applied according to the value of maximum crop evapotranspiration (ETc) (Penman–Monteith equation): a first treatment applying 100% ETc, corresponding to being fully irrigated; a second treatment applying 50% ETc; and a third treatment in which neither irrigation nor rain inputs occurred (non-irrigated) from the end of the pit hardening period until harvest. The experiment was repeated in two years, during the 2018/2019 and 2020/2021 seasons, with a different randomization of the experimental units. The assays were specifically made in years of high fruit load.

### 2.2. Soil, Irrigation and Tree Water Status

The soil at this site has a fine textured A horizon, with a maximum depth of 30 cm, 2.9% organic matter and pH 6.3, corresponding to a Typic-Vertic Argiudolls soil according to the USDA classification [22]. The soil water curve retention was characterized by measuring water content at tensions of 0.01 and 1.5 MPa (field capacity and permanent wilting point, respectively), using the Richards and Weaver methods [23]. Undisturbed soil sample were used for soil water extraction from different depths up to 0.50 m. Soil moisture was monitored throughout the experiment using three FDR sensors installed at three different depths (15, 30 and 45 cm deep), using an EM50 Digital/Analog Data Logger (Decagon Devices, Inc., Pullman, WA, USA). The total amount of applied water was 190 mm and 410 mm in the first season, and 240 mm and 540 mm in the second season, for 50% ETc and

100% ETc, respectively. Prior to the installation of the experiment, the crop was irrigated according to the value of maximum crop evapotranspiration, so that once the treatments were started, the soil was at field capacity. To avoid the effect of rainfall, after pit hardening the soil around the trees (24 m$^2$/tree) was covered with nylon (bilayer) in all treatments. The plastic was placed to prevent the entry of rain into the soil and it was not removed until harvest, so that the rain from January to May was not available for the plants and therefore did not affect the treatments. A complete drip irrigation system was used to supply the irrigation water. The system consisted of a 16-diameter (13.6 mm) lateral pipe PE (0.25 MPa) with 0.20 m of emitter spacing. The flow of the self-compensating emitters was 4 L h$^{-1}$. Therefore, the system was designed to apply 7 mm h$^{-1}$ of water with an average pressure in the lateral pipe of 100 kPa.

The irrigation schedule for the 100% ETc treatment was accomplished daily using the simplified water balance method for the root zone of the crop [24,25], according to the following Equation (1):

$$Dri = ETci - Pei - Ii + DPi + Dri - 1 \tag{1}$$

where *Dr* stands for root zone depletion (mm); *ETc* for maximum crop evapotranspiration (mm), computed as $ET_0$*Kc*; *Pe* for effective precipitation (mm); *I* for irrigation depth (mm); *DP* for deep percolation outside the root zone (mm); *i* for the current day; and *i* − 1 for the day before. The value used for *Kc* was 0.65 at the beginning of the season and then 0.70 during the mid-season and end-season stages.

The potential crop evapotranspiration was determined as $ET_0 \times kc$ (*kc* values used to calculate the water balance, according to [26]). The irrigation schedule for the 50% ETc treatment was carried out using the same methodology as used for the 100% ETc treatment. The effective precipitation (Pe) used in the soil water balance equation was 0 during pit hardening until harvest. Irrigation depth was computed so that the depletion-water root zone was between the field capacity and readily available water [24]. The daily water balance was calculated for each irrigation treatment with the FAO 56 method [27], and it was used to apply the irrigation during both seasons.

Tree water status was assessed by measuring the stem water potential (SWP) using a Scholander-type pressure chamber (Soil Moisture Equipment Corp., Santa Barbara, CA, USA) every 15 days from the end of pit hardening until harvest. Two hours before the measurement, the shoot was enclosed in a plastic bag, allowing the leaf water potential to balance with the stem water potential for a more stable value than that of the leaf water potential. Measurements on each tree were made between 12h00 and 14h00 on mature leaves exposed to the sun from the middle of the branches [28,29]. The measured SWP values were accumulated over the irrigation period and the cumulative leaf water potential (CLWP) was calculated to compare the level of water stress throughout the entire experiment [30].

### 2.3. Climate

The climate is temperate humid with an average annual rainfall of 1200 mm unequally distributed throughout the year. The rainfall, mean air temperature, relative air humidity, total day radiation and wind speed were obtained from a Campbell automatic weather station located near the experiment (approximately 0.5 km) (Figure 1). Considering the average historical data (1974–2020), the $ET_0$ values are as follows: 456.3 mm in summer (maximum values in December and January), 205.2 mm in autumn, 99 mm in winter (minimum values in June and July) and 310.5 mm in spring, with an annual total of 1071 mm.

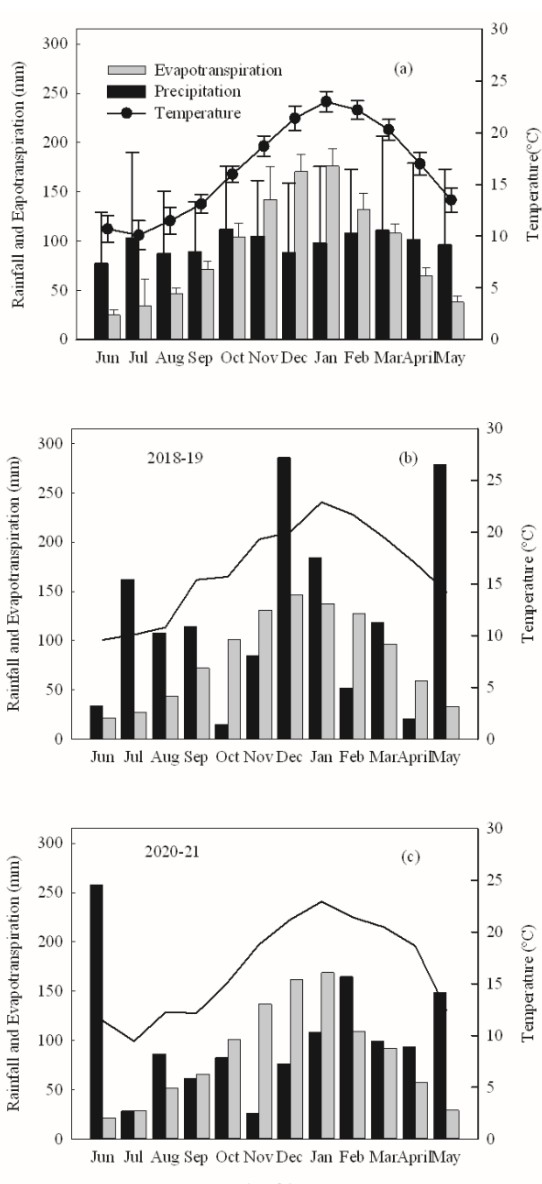

**Figure 1.** Average monthly values of mean air temperature (continuous line) (°C), evapotranspiration (gray bars) ($ET_0$, mm × 10) and precipitation (black bars) (mm) at the experimental site in INIA Las Brujas from 1973 to 2018 (**a**), for the 2018/2019 season (**b**), and for the 2020/2021 season (**c**). Vertical bars in (**a**) indicate the standard deviation. Data were recorded at an automatic weather station and is available at http://www.inia.uy/gras/Clima/Banco-datos-agroclimatico 11 January 2022.

Historical data for precipitation and $ET_0$ show that during spring and summer evapotranspiration exceeds precipitation (Figure 1a), which causes a water deficit of approximately 250 mm during that period. The two seasons in which the experiments were carried out showed differences in rainfall and evapotranspiration. In the first season, high precipitations during December and January were recorded, which by far exceeded the evapotranspiration, and a water deficit occurred in October, November and February (Figure 1b). This generated differences in temperature and relative humidity and, consequently, a vapor pressure deficit that was lower compared to the second season of the experiment during January and higher during February (Figure 2). In the second season (Figure 1c), evapotranspiration exceeded rainfall by approximately 300 mm between the months of September and February, after which the values recorded were similar to what has occurred historically (Figure 1a). In order to characterize our climate region, we compared the VPD

of Uruguay and Spain (Córdoba), a main traditional olive cultivation region, considering the average temperature (24 h) and average relative humidity for the 2009–2020 (Uruguay) and 2001–2020 (Spain) periods (Supplementary Figure S1).

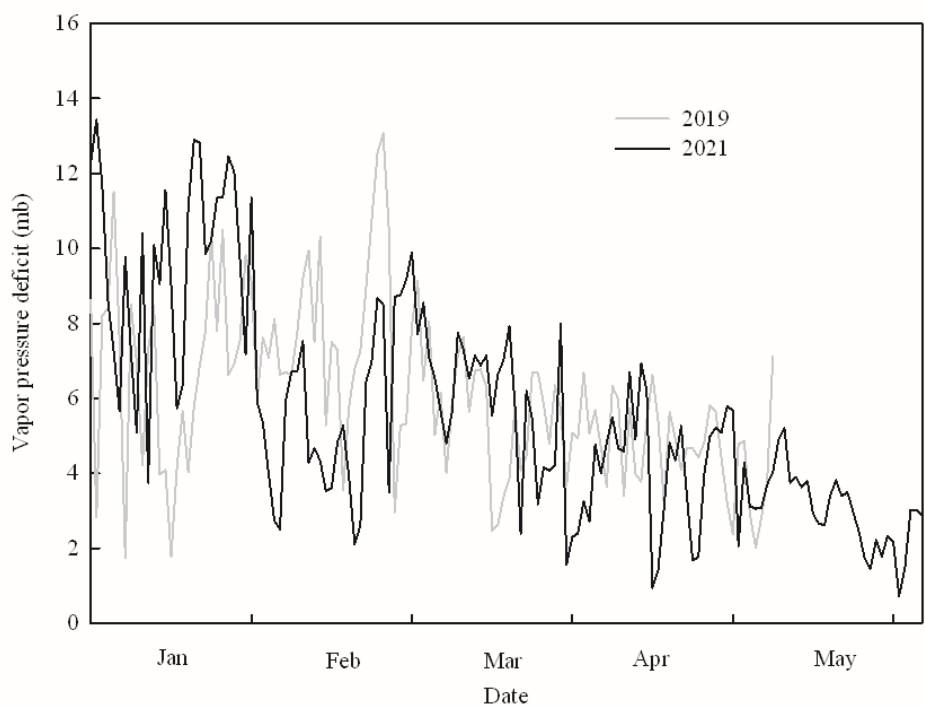

**Figure 2.** Daily vapor pressure deficit (mb) in Uruguay for the 2019 and 2021 seasons from 1 January to 30 April. Data were recorded at an automatic weather station at the experimental site in INIA Las Brujas (available at http://www.inia.uy/gras/Clima/Banco-datos-agroclimatico 11 January 2022).

### 2.4. Productive Parameters

The fresh weight of the fruit (g), pit weight (g), and pulp/pit ratio were recorded monthly in 20 olive fruits per tree from the end of pit hardening to harvesting time. Each tree was harvested individually with a trunk vibrator machine at the end of April in both seasons, and fruit maturity index (MI) was recorded based on a 0–7 scale [31]. Fruit yield (kg/tree) was recorded and fruit number per tree was calculated from fruit yield and mean fresh weight of the fruit.

Oil content (%) was measured monthly from the end of pit hardening to harvest time on a sample of 200 g of olives per tree. To determine the fruit moisture content, each sample was ground with a hammer mill and dried at 105 °C for 48 h, after which the dried sample was grinded with a mortar and the oil content was determined following the Soxhlet method [32]. Olive oil from each tree was obtained in an Abencor mill (Mc2 Ingenieria y Sistemas, Sevilla, Spain) for oil composition analysis. Water productivity (WPf) was calculated as kg of fruit per unit of water applied ($m^3$) throughout the experiment [14].

### 2.5. Oil Chemical Composition

Acidity: the free fatty acid content was determined following the official analytical method described in [33] and expressed as acidity percentage.

Oil pigments: the content of total chlorophylls and total carotenoids was determined in a spectrophotometer by dissolving 7.5 g of each oil in cyclohexane and measured at 670 and 470 nm, respectively, according to Minguez-Mosquera et al. [34], and expressed as mg kg$^{-1}$ EVOO.

Fatty acids profile: fatty acid methyl esters were prepared from trans-esterification reactions with a cold methanolic solution of potassium hydroxide and analyzed by gas chromatography as described by Feippe et al. [35], with some modifications. Briefly, 0.1 g

of the olive oil sample were dissolved in 2 mL of heptane and vortexed for 1 min at 20 °C, then centrifuged at 12,000 rpm for 5 min. The supernatant was trans-esterified by adding 1 mL 4 N KOH in methanol and stirring manually for 1 min. The solution was dried with sodium sulfate powder, centrifugated at 12,000 rpm for 5 min, filtered and injected into a gas chromatograph with a flame ionization detector (GC-FID, Shimadzu model 2010-Plus, Kyoto, Japan). The column used was an Agilent DB-WAX (30 m × 0.25 mm ID, 0.25 μm). The injection temperature was 250 °C and the FID detector temperature was set at 300 °C. The oven temperature was set at 160 °C, increased to 200 °C after 13 min and maintained for 22 min, after which the temperature was increased to 240 °C for the final 25 min. The sample injection volume was 1 μL, and the mobile phase used was nitrogen at 30 mL/min. The hydrogen flow was set at 40 mL/min and the air flow at 400 mL/min. A standard certified Fatty Acids Methyl Esters (FAME) mix (Sigma-Aldrich, St Louis, MO, USA) was used to identify the peaks according to retention times and expressed as percentages.

Total phenolics in olive fruits: total phenolics (TP) were determined according to the method adapted from Sánchez-Rangel et al. [36] with a Folin–Ciocalteu reagent. Two grams of olives were homogenized in an Ultraturrax for 2 min, extracted with 10 mL of 80% methanol, and centrifuged at 10,000 rpm for 4 min at 4 °C. The TP determination was carried out in 96-well microplates, with gallic acid (GA) as the calibration standard; 15 μL of diluted sample extract or GA dilutions were incubated for 15 min in the dark, after the addition of 240 μL of distilled water, 15 μL of Folin–Ciocalteu reagent and 30 μL of 1 N sodium carbonate. The absorbance was read at 760 nm in a Synergy H1 Hybrid Multi-Mode Reader with Gen 5 software (Bio-Tek Inc., Winooski, VT, USA). The results were expressed as mg of gallic acid equivalents (GAE) per kg of fresh olives.

Total phenolic analysis in Virgin Olive Oils (VOO): total phenol content was determined by the Folin–Ciocalteu method described by Gutfinger [37], with some modifications. Briefly, 5 g of olive oil samples were dissolved in 7 mL of MeOH:$H_2O$ (80:20) and vortexed. The mixture was centrifuged for 10 min at 5800 rpm and the procedure was repeated twice. The supernatants were pooled and brought up to a volume of 25 mL with MeOH:$H_2O$ (80:20). An aliquot of 1 mL was transferred to a 10-mL volumetric flask to which 8 mL of distilled water were added followed by 0.5 mL of Folin–Ciocalteu reagent and 0.5 mL of saturated $Na_2CO_3$. The samples were shaken and left for 15 min in the dark at room temperature. The absorbance was determined spectrophotometrically at 760 nm in a UV–VIS spectrophotometer (Shimadzu™ model UV-3000, Kyoto, Japan). The total amount of TP was calculated and expressed as mg of GAE equivalent per kg of oil by using a calibration curve prepared with pure gallic acid standard solution.

### 2.6. Statistical Analysis

Since fruit number is a yield component defined during fruit set [38–40] and irrigation treatments are installed after that phase, we analyzed whether there were significant differences between the treatments of fruit number per tree at harvest. As significant differences were detected, productive variables were analyzed with ANCOVA, with fruit number per tree as a covariate. The adjusted model for each cultivar included treatment effect, year effect, and their interaction. The Mixed Models procedure (SAS v.9.4 (SAS Institute, Cary, NC, USA 2013) was used, and the corrected means were contrasted using the Tukey–Kramer test, with a significance level of 5%. Linear functions were fitted to the relationships between the fresh weight of the fruit, fresh weight of the pulp, pulp/pit ratio, maturity index and the CLWP variables. We report those functions that provided the best fits with a significance level of 5%.

## 3. Results

### 3.1. Tree Water Status and Fruit Moisture

Plants water status was affected by the levels of irrigation applied during the experiment. The ranges of stem water potential (SWP) during the experiment in both seasons and for both cultivars are presented in Table 1. The values obtained for the non-irrigated

treatment were lower than those for irrigated trees. The most negative water potential value reached throughout the experiment was −3.5 MPa, corresponding to the non-irrigated treatment. In the 2021 season, water potential values were more negative than in the 2019 season. During spring and summer of 2021 there were few rain events. The estimated evapotranspiration during those months was higher than the precipitation. The crop water demand exceeded the water supply from the rains. Despite the differences in the ranges between the two seasons, the differences between the treatments in each season were clearly defined and had the same response pattern.

**Table 1.** Midday stem water potentials (Ψstem) ranges obtained from Arbequina and Frantoio cultivars grown under different water irrigation levels (non-irrigated, 50% ETc and 100% ETc) in two seasons.

| Cultivar | 2019 | | | 2021 | | |
|---|---|---|---|---|---|---|
| | Non-Irrigated | 50% ETc | 100% ETc | Non-Irrigated | 50% ETc | 100% ETc |
| cv. Arbequina | −1.7 to −2.3 | −1.3 to −1.6 | −0.8 to −1.3 | −2.5 to −3.4 | −1.7 to −2.1 | −1.3 to −1.8 |
| cv. Frantoio | −1.9 to −2.7 | −1.5 to −1.6 | −1.1 to −1.5 | −3.1 to −3.5 | −2.0 to −2.8 | −1.9 to −2.4 |

The values of cumulative leaf water potential (CLWP) of the non-irrigated treatment were lower than those of irrigated trees in both cultivars and seasons (Figure 3). The results show that the generated water deficit was progressive and constant throughout the experiment. The water deficit level reached in the 2021 season was more intense than in 2019 in both cultivars, 'Frantoio' being the cultivar that reached the most negative values. In the 2019 season, values of −248 and −295 MPa in Arbequina and Frantoio, respectively, were recorded in the non-irrigated treatments, while the 100% ETc treatment presented values between −137 and −149 MPa. In the 2021 season, values of −353 and −403 MPa in Arbequina and Frantoio, respectively, were recorded in non-irrigated treatments while the 100% ETc treatment presented values between −211 and −252 MPa.

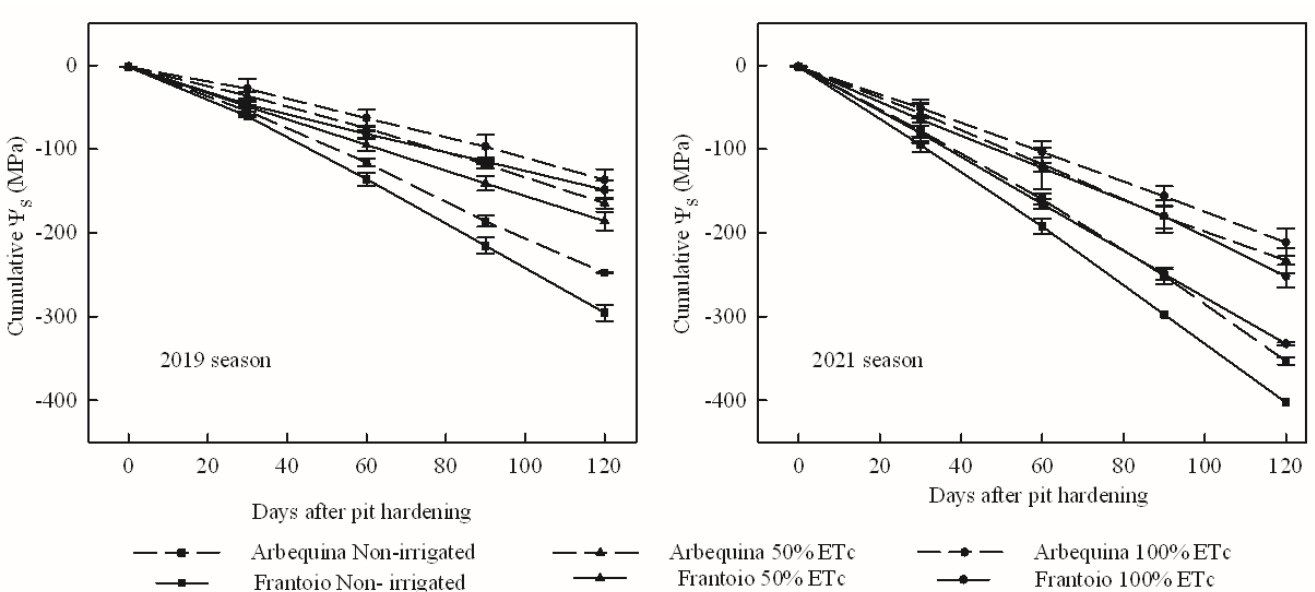

**Figure 3.** Seasonal evolution of cumulative leaf water potential (CLWP, MPa) of Arbequina (dotted line) and Frantoio (solid line) cultivars from the end of pit hardening to harvest time in the 2019 and 2021 seasons. Treatments included non-irrigated (■), 50% ETc (▲) and 100% ETc (●). Values are the means of four trees.

Fruit moisture content recorded during the experiment also presented differences between treatments, being lower in non-irrigated treatments than in irrigated ones, for both cultivars and both seasons (Figure 4). At the end of the experiment, the 100% ETc treatment

presented at least 12% more moisture than the non-irrigated treatment in Arbequina and more than 8% in Frantoio. In 2019, fruit moisture in the non-irrigated treatments was 43 and 48% in Arbequina and Frantoio, respectively, compared to the respective 59 and 56% recorded in the 100% ETc treatments. In 2021, fruit moisture in the non-irrigated treatments was 54 and 49% in Arbequina and Frantoio, respectively, compared to the respective 66 and 63% recorded in the 100% ETc treatments.

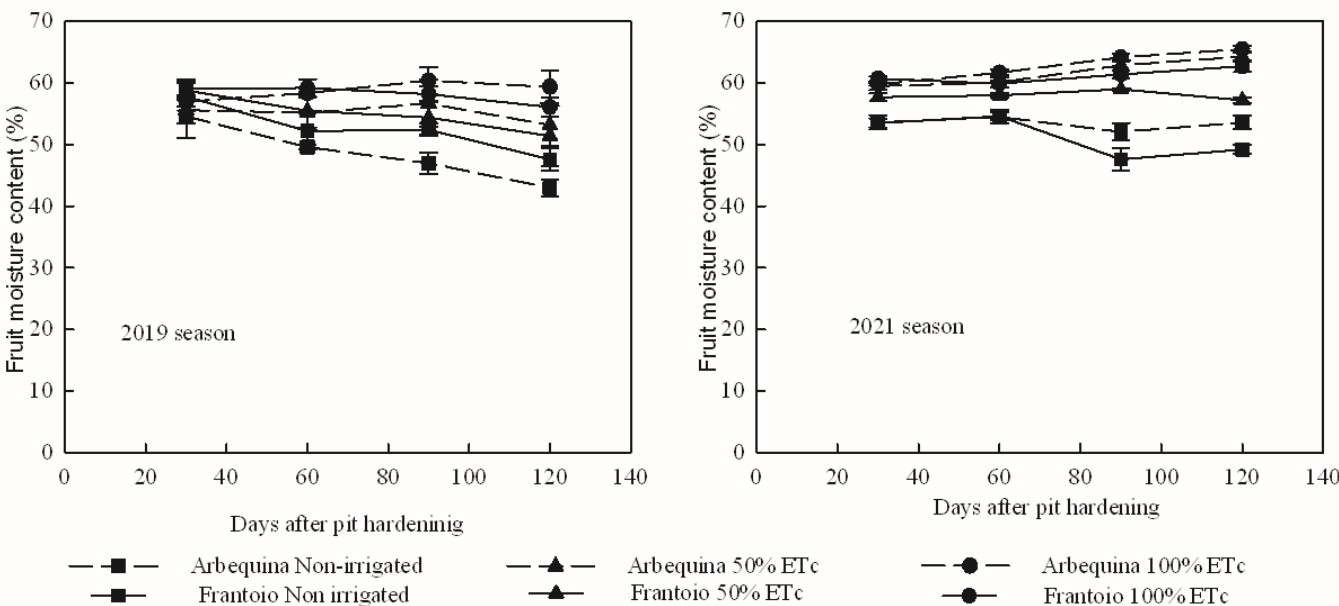

**Figure 4.** Fruit moisture content (%) in Arbequina (dotted line) and Frantoio (solid line) cultivars from the end of pit hardening to harvest time in the 2019 and 2021 seasons. Treatments included non-irrigated (■), 50% ETc (▲) and 100% ETc (●). Values are the means of four trees.

*3.2. Productive Parameters*

Significant differences between the treatments of fruit number per tree at harvest were detected, so productive variables were analyzed with ANCOVA, with fruit number per tree as a covariate. Fruit yield (kg/tree) did not show significant differences in Arbequina. In Frantoio, however, fruit yield was significantly higher in the irrigated treatments than in the non-irrigated treatment (Table 2). The maturity index for both cultivars was higher in the non-irrigated treatments than in the irrigated ones. Oil content (% DWB) did not show significant differences between treatments in Arbequina, whereas in Frantoio differences were only observed when comparing the 100% ETc and the non-irrigated treatment, with the oil content being higher in the latter. The fruit yield achieved in both irrigated treatments was similar. Therefore, regardless of the cultivar, WPf was higher in the treatment with 50% ETc than in the one with 100% ETc (Table 2). WPf was not calculated in the non-irrigated treatment, since there was no application of irrigation.

Correlations between the production parameters and CLWP were analyzed (Figure 5). CLWP better represents plant water status when it is closer to zero. Fresh weight of fruit increased with the best water status in both cultivars and in both seasons, as did the fresh weight of the pulp and pulp/pit ratio, presenting significant regressions in all cases. A positive relationship was observed between the fresh weight of fruit and CLWP, as irrigated trees of both cultivars presented a higher fresh weight of fruit than those of the non-irrigated treatment (Figure 5). In Arbequina, an adjustment greater than 0.4 was obtained, while in Frantoio this value was greater than 0.66. Similar results were obtained in the relationship between fresh weight of pulp and CLWP, with an adjustment greater than 0.51 in Arbequina and at 0.65 in Frantoio. A positive relationship was also observed between the pulp/pit ratio and CLWP, as Arbequina and Frantoio exhibited an adjustment greater than 0.68

and 0.71, respectively. The maturity index presented a negative relationship with CLWP, being significantly higher in non-irrigated treatments compared to those with irrigation (Figure 5). Arbequina presented an adjustment of 0.72, with MI ranges that varied between 1 and 3.7, while Frantoio presented an adjustment of 0.40, with MI ranges between 1.1 and 2.1 (Table 2).

**Table 2.** Fruit yield (kg/tree), maturity index, oil content (%) and water productivity (WPf, kg fruit/m$^3$ water applied) of Arbequina and Frantoio cultivars grown in the fully irrigated treatment with 100% ETc, in treatment with 50% ETc and in the non-irrigated treatment at INIA Las Brujas. Mean of the two seasons.

| Evaluated Parameters | cv. Arbequina | | | cv. Frantoio | | |
|---|---|---|---|---|---|---|
| | Irrigation Treatment | | | | | |
| | Non-Irrigated | 50% ETc | 100% ETc | Non-Irrigated | 50% ETc | 100% ETc |
| Fruit yield (kg/tree) | 35.2 [a,*] | 42.2 [a] | 45.2 [a] | 31.5 [b] | 45.5 [a] | 52.4 [a] |
| Maturity index | 3.32 [a] | 2.21 [b] | 1.91 [b] | 2.31 [a] | 1.32 [b] | 0.96 [b] |
| Oil content (% DWB) | 39.6 [a] | 37.7 [a] | 37.6 [a] | 39.2 [a] | 36.9 [a,b] | 34.5 [b] |
| WPf (kg fruit/m$^3$ water applied) | [#] | 19.6 | 9.5 | [#] | 21.2 | 11.0 |

\* Different letters within the row indicate significant differences for each cultivar separately (HSD Tukey–Kramer $p \leq 0.05$). [#] Since the non-irrigated treatment did not receive water applications, the WPf was not calculated.

### 3.3. Polyphenols Content in Fruits

Total phenols in fruit showed a negative relationship with the reduction in CLWP (Figure 6). Arbequina showed a reduction of 2730 and 1180 mg GAE/kg FW in total phenols in fruit, as CLWP was lower in 2019 and 2021, respectively. Frantoio showed a lesser reduction, being 470 and 610 mg GAE/kg FW in total phenols in fruit, as CLWP was lower in 2019 and 2021, respectively.

### 3.4. Oil Chemical Composition

The fatty acids profile was affected by irrigated treatments in both seasons, mainly for Arbequina. In the 2019 season, palmitoleic (C16:1) and linoleic (C18:2) acids increased in the non-irrigated treatment, whereas stearic (C18:0) and oleic (C18:1) decreased in Arbequina. In Frantoio, only stearic acid showed an increase without irrigation. In the 2021 season, Arbequina showed higher levels of linoleic acid in the non-irrigated treatment, similarly to the first season but in a much higher percentage in all treatments. Arachidic (C20:0) and eicosenoic (C20:1) acids showed lower levels under non-irrigated treatments in the 2021 season in Arbequina. As for Frantoio, linolenic (C18:3) and eicosenoic acids were lower and stearic acid was higher in the non-irrigated treatment. The MUFA/PUFA ratio was modified only in Arbequina in the first season, being significantly higher in the 100% ETc treatment. The oil polyphenol content was significantly higher in the non-irrigated treatment than in the 100% ETc treatment for Arbequina in the 2019 season and for Frantoio in 2021. Total carotenoids were significantly higher in the non-irrigated treatments in both cultivars, except for Arbequina in 2021, when no significant differences were recorded. Total chlorophylls in the non-irrigated treatment were lower in Arbequina in both seasons and higher in Frantoio, which only presented significant differences with the other treatments in the 2021 season. Free acidity was analyzed as the quality control of the extraction process, which in all cases was less than 0.15% (data not shown).

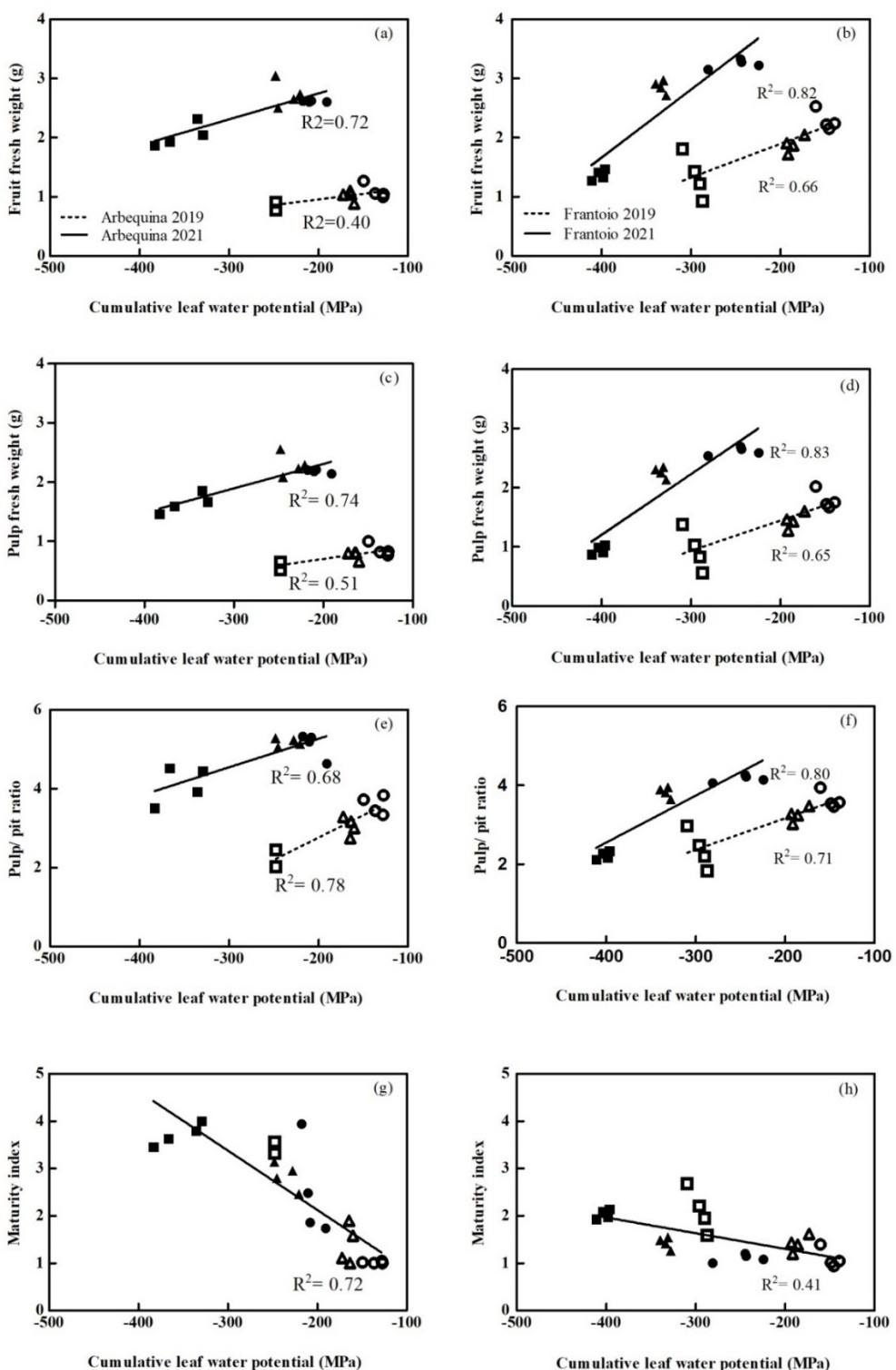

**Figure 5.** Fresh weight of fruit in grams (**a**,**b**), fresh weight of pulp in grams (**c**,**d**), pulp/pit ratio (in fresh weight basis) (**e**,**f**) and maturity index (**g**,**h**) at harvest time as a function of the cumulative leaf water potential (MPa) in Arbequina (**left**) and Frantoio (**right**) cultivars. Treatments included: non-irrigated (■), 50% ETc (▲) and 100% ETc (●). In panels (g) and (h), the regression was done for the two years together. The empty symbols correspond to the 2019 season and the full symbols to 2021.

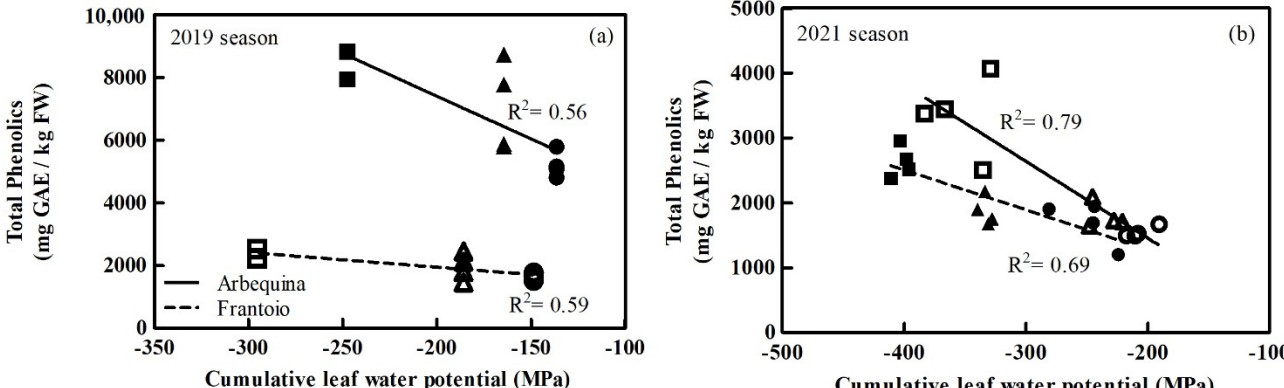

**Figure 6.** Total phenolics in fruit (mg GAE/kg FW) as a function of the cumulative leaf water potential (MPa) in the 2019 (**a**) and 2021 (**b**) seasons. Treatments included non-irrigated (■), 50% ETc (▲) and 100% ETc (●). Full symbols correspond to cv. Arbequina and empty symbols to cv. Frantoio.

## 4. Discussion

Olive (*Olea europaea* L.) is a typical tree of the Mediterranean climate that has traditionally been cultivated in rainfed conditions and in regions with high vapor pressure deficit (VPD). VPD is the difference between the saturation vapor pressure and real vapor pressure during a given period [27]. Mairech et al. [5] express the need for calibrating irrigation at a local level given the high dependence of the water requirements with VPD. Many authors report the range of xylem potential reached during irrigation experiments, but few studies refer to VPD [15,41]. We compared the VPD of Uruguay and Spain (Córdoba) (Supplementary Figure S1) and it is observed that Uruguay presents notoriously lower values of VPD. During the summer, VPD values in Spain double those of Uruguay. VPD is directly associated with evapotranspiration ($ET_0$), as higher VPD would generate higher water consumption. Annual $ET_0$ values in our conditions are lower than those reported for other olive regions around the world [1,5]. During summer, the period in which the experiment was carried out, the most frequent $ET_0$ value in Uruguay was 456 mm (Figure 1a), while in the Mediterranean basin it is higher (for instance, 600 mm in Sevilla [1]). These climatic differences can influence olive tree physiology, affecting the productive variables.

Plant water status achieved by the different treatments in both seasons generated a moderate stress in the 50% ETc treatment and a severe one in the non-irrigated treatment according to Fernández et al. [42]. Stem water potential (SWP) in both seasons was in the range of −1.7 and −3.5 MPa in the non-irrigated treatment, −1.3 and −2.8 MPa in the 50% ETc treatment and between −0.8 and −2.4 MPa the in 100% ETc treatment, all within the range measured by other authors [14,30,43]. In Arbequina, the stress during the experiment reached cumulative leaf water potential gradients between treatments from −137 to −248 in 2019 and from −211 to −353 in 2021. In Frantoio, the CLWP gradient was −149 to −295 in 2019 and −251 to −403 in 2021 (Figure 3). These values were similar to those reported by Gucci et al. [30]. This allows us to classify the stress level of treatments and therefore to compare our results with reports made in other sites and cultivars.

Final yield is determined by fruit number, fruit weight and oil content [44]. Fruit number is defined in the flowering–fruit setting phase, prior to establishing the experiment [38–40]. As significant differences in fruit number per tree were detected at harvest between treatments, this parameter was used as a covariate. Therefore, we focused on fruit weight and oil content responses. It was observed that, under the experimental conditions, both cultivars responded positively to irrigation, increasing the fruit weight and the pulp/pit ratio in comparison to the non-irrigated treatment. A similar response was also observed during the same water deficit period by Lavee et al. [45] in cv. Muhasan, where fruits were significantly smaller in the non-irrigated trees than in the irrigated ones. The magnitude of the irrigation effects was different depending on the cultivar. Fruit weight in

Arbequina in the irrigated treatment increased by 40% in comparison to the non-irrigated treatment, while in Frantoio it increased by 68%, which represents a 66% increase in yield per tree. Moreover, a positive linear relationship between the pulp/pit ratio and water plant status was observed in both cultivars (Figure 5b). This is in accordance with previous studies by Gómez-Rico et al. [46] and Lavee et al. [45] who observed that irrigation increases the pulp/pit ratio in comparison to rain-fed trees. However, other authors did not find effects in this ratio under mild water stress [12,45]. In the framework of this experiment, the impact of irrigation on absolute fruit weight was lower in Arbequina, a cultivar with small fruits. In addition, it did not translate into a yield increase (kg/plant). In Frantoio, the increase in fruit weight doubled with irrigation, impacting the yield (Table 2 and Figure 5). This could be due to the genetic characteristic of the fruit size limiting the response to irrigation.

Fruit oil content does not yet show a consensual pattern of response to water restriction. This parameter is highly variable depending on the moment and level of water restriction applied [14,46]. In Arbequina, we found no effect of irrigation on the oil content on a dry basis (Table 2). Similar results were obtained by Hueso et al. [14] in Arbequina under similar water-stress conditions (up to −2.6 MPa) and by Ahumada-Orellana et al. [47] even under severe water stress (up to −6 MPa). However, Iniesta et al. [40] found a higher oil content in water deficit treatments in comparison to irrigated ones, under similar stress conditions as those of our work (−2.9 and −3.6 MPa), also in Arbequina. Oil content is conditioned by the genotype–environment interaction [48]. In this study, a different response in oil content according to cultivars was observed, in agreement with Iniesta et al. [40]. In particular, the oil content in Frantoio was higher in the non-irrigated treatment than in the 100% ETc treatment (Table 2), while in other works a reduction in oil content has been recorded when stem water potential was near −4 MPa [46,49].

Regarding MI, the negative effect of the fruit load on the advancement of maturity has been widely reported. In this sense, it is expected that the treatments with more load have a delayed maturity. Despite having corrected the maturity mean values for fruit number, a negative linear relationship was maintained between MI and water status in both cultivars, with a greater slope in Arbequina (Figure 5). The same pattern was found by Inglese et al. [50] in cv. Carolea when the irrigation in the final phase of fruit development delayed the MI, and by Motilva et al. [51] in Arbequina. However, Iniesta et al. [40] did not find that a deficit in irrigation leads to an earlier ripeness.

Polyphenols have been associated with defense mechanisms against water stress [52]. In our study, total phenols in fruit increased in the non-irrigated treatment in both cultivars. According to Talhaoui et al. [53], the transfer of phenolic compounds from fruits to oil did not surpass 2% in a study with six cultivars that explains qualitative and quantitative changes in phenolic compounds of olive oil during oil extraction in relation to fruits. Our results show that the content of polyphenols in Arbequina and Frantoio oil increased for the non-irrigated treatment compared with the 100% ETc treatment, as also reported by several authors [51,54,55]. During the 2021 season, Arbequina showed the same tendency but the differences were not statistically significant. In the 2019 season, the content of polyphenols in Frantoio increased with irrigation restriction at 50% ETc, reaching the highest level. A similar effect was observed by Tognetti et al. [56], who reported that total phenolics at 66% ETc where higher than in the non-irrigated and 100% ETc treatments. Other works with Frantoio found inconsistent results between years in the phenol content in response to the level of irrigation [57]. Despite the fact the levels found in both seasons are different, they are in concordance with previously reported oils from similar experiments [18,46,51]. In humid climate conditions, the differences between two growing seasons could affect not only the oil content [58] but also the minor oil components, such as phenolic compounds and the fatty acid profile.

The carotenoids content in oil was affected by plant water status, increasing with water restriction between the non-irrigated and 100% ETc treatments in Arbequina and Frantoio, respectively. Only Arbequina showed no differences in season 2021 (Table 3). However, previous works on the effects of irrigation on the carotenoid content found

different responses. Sena-Moreno et al. [59] reported an increase in carotenoids with water restriction, consistent with our results, while Tovar et al. [18] did not find differences. The chlorophyll content in Arbequina was reduced by 65% (2019) and 81% (2021) in the non-irrigated treatment compared to the irrigated ones. This response was probably due to the advanced maturity index in the non-irrigated treatment at harvest. For Frantoio, the only difference was found in the 2021 season, where the chlorophyll content increased by 76%. The pigments content (chlorophylls and carotenoids) varies depending on the cultivar, the fruit ripening stage, the weather conditions and the oil-extraction processes [60]. The lipophilic characteristics of these compounds determine the affinity of the oily phase and the migration ratio into the EVOO, playing a role in oxidative stability [61].

**Table 3.** Physicochemical composition of olive oil in the non-irrigated, 50% ETc and 100% ETc treatments: total phenolics, chlorophylls, total carotenoids, free acidity and fatty acids composition. Olives from the Arbequina and Frantoio cultivars harvested in 2019 and 2021 obtained the corresponding virgin olive oil (VOO).

| | 2019 | | | | | | 2021 | | | | | |
|---|---|---|---|---|---|---|---|---|---|---|---|---|
| | cv. Arbequina | | | cv. Frantoio | | | cv. Arbequina | | | cv. Frantoio | | |
| | Non-Irrigated | 50% ETc | 100% ETc | Non-Irrigated | 50% ETc | 100% ETc | Non-Irrigated | 50% ETc | 100% ETc | Non-Irrigated | 50% ETc | 100% ETc |
| Total Phenolics (mg GAE/kg EVOO) | 147.5 [a],* | 138.3 [a,b] | 121.7 [b] | 343.5 [a,b] | 372.4 [a] | 306.4 [b] | 86.4 [a] | 72.6 [a] | 80.2 [a] | 133.5 [a] | 105.9 [b] | 104.0 [b] |
| Totals Carotenoids (mg Car/kg EVOO) | 3.54 [a] | 3.36 [b] | 2.88 [b] | 7.29 [a] | 5.81 [b] | 5.59 [b] | 0.68 [a] | 0.77 [a] | 0.79 [a] | 5.19 [a] | 3.08 [b] | 3.41 [b] |
| Total Chlorophylls (mg Ch/kg EVOO) | 0.69 [b] | 1.94 [a] | 1.99 [a] | 6.81 [a] | 5.16 [a] | 5.79 [a] | 0.14 [b] | 0.75 [a] | 0.75 [a] | 7.47 [a] | 3.85 [b] | 4.04 [b] |
| Fatty acid composition: | | | | | | | | | | | | |
| Palmitic Acid (%) | 14. 82 [a] | 15.07 [a] | 15.25 [a] | 13.54 [a] | 14.19 [a] | 13.67 [a] | 16.69 [b] | 17.41 [a] | 17.18 [a,b] | 14.00 [a] | 13.81 [a] | 14.64 [a] |
| Palmitoleic Acid (%) | 2.19 [a] | 1.99 [a,b] | 1.93 [b] | 1. 23 [a] | 1.45 [a] | 1.55 [a] | 1.62 [b] | 2.16 [a,b] | 2.28 [a] | 1.07 [a] | 0.85 [a] | 1.09 [a] |
| Stearic Acid (%) | 1.51 [c] | 1.74 [a] | 1.65 [b] | 2.25 [a] | 1.89 [b] | 1.58 [b] | 1.71 [a] | 1.75 [a] | 1.69 [a] | 2.26 [a] | 1.90 [b] | 1.70 [c] |
| Oleic Acid (%) | 67.56 [b] | 68.73 [a,b] | 69. 57 [a] | 73.45 [a] | 73.05 [a] | 74.07 [a] | 60.23 [a] | 60.52 [a] | 61.24 [a] | 71.10 [a] | 70.28 [a] | 71.01 [a] |
| Linoleic Acid (%) | 12.84 [a] | 11.41 [b] | 10.42 [c] | 8.17 [a] | 8.14 [a] | 7.78 [a] | 18.42 [a] | 16.66 [a,b] | 16.14 [b] | 10.11 [b] | 11.59 [a] | 9.808 [b] |
| Linolenic Acid (%) | 0.50 [a] | 0.51 [a] | 0.56 [a] | 0.75 [a] | 0.68 [b] | 0.68 [b] | 0.60 [a] | 0.65 [a] | 0.66 [a] | 0.63 [c] | 0.70 [b] | 0.82 [a] |
| Arachidic Acid (%) | 0.30 [a] | 0.29 [a] | 0.31 [a] | 0.31 [a] | 0.31 [a] | 0.31 [a] | 0.34 [a] | 0.40 [a] | 0.40 [a] | 0.41 [a] | 0.40 [a] | 0.40 [a] |
| Eicosenoic Acid (%) | 0.28 [a] | 0.27 [a] | 0.31 [a] | 0. 30 [a] | 0.29 [a] | 0.37 [a] | 0.27 [b] | 0.31 [a] | 0.30 [a] | 0.31 [c] | 0.36 [b] | 0.40 [a] |
| MUFA/PUFA | 5.23 [c] | 5.94 [b] | 6.55 [a] | 8.39 [a] | 8.49 [a] | 9.00 [a] | 3.28 [a] | 3.66 [a] | 3.81 [a] | 6.76 [a] | 5.84 [b] | 6.83 [a] |

* Means ($n = 4$) followed by the same letter within a row for each cultivar and each season are not significantly different at $p < 0.05$ (Tukey's test). GAE: Gallic acid equivalent; Ch: total chlorophylls; Car: total carotenoids. MUFA/PUFA: $\sum$ monounsaturated fatty acids (MUFA)/$\sum$ polyunsaturated fatty acids (PUFA) ratios.

The effect of different irrigation strategies on the olive oil fatty acid composition remains unclear [4]. In our study, oleic acid, the main olive oil fatty acid, was significantly lower for Arbequina in the 2019 non-irrigated treatment (Table 3). Severe and prolonged water stress during fruit growth increases fruit temperature [62] and consequently the oleic acid proportion fell [63]. In the same regard, we found a reduction in oleic acid as linoleic acid increases with severe water restriction in Arbequina. The MUFA/PUFA ratio decreased concomitantly for Arbequina in the 2019 season in the non-irrigated treatment, with no changes in Frantoio in either season. Despite slight differences in the percentages, the obtained quality complied with the IOC specifications for EVOOs regarding the fatty acid composition of both cultivars.

It is important to find the best balance between yield, oil quality and water-saving issues [30]. The relationship between fruit production and ETc was shown to be curvilinear by Moriana et al. [10], which means that high production could be reached at lower values than those of the maximum potential ETc. It was demonstrated that under full irrigation olives can achieve high yields (8 t/ha/year) in humid template regions [58]. In our study, if we compare WP (estimated according to kg/tree based on the applied water) between both irrigation treatments, we observe that it was always higher in the 50% ETc treatment than in the 100% ETc one, since yields in kg/tree were similar between both and the water applied was half in the 50% ETc one. Moreover, oil polyphenols content was similar in both irrigated treatments, which may affect oil stability [64]. Fruit moisture at harvest ranged between 43% and 65.5% in Arbequina in the non-irrigated and 100% ETc treatments, respectively. In Frantoio, the range was between 47.6% and 62.7% in the non-irrigated and 100% ETc treatments, respectively. High fruit moisture may have negative effects on the

oil extraction process [65,66]. A reduction in irrigation translates into a lower percentage of fruit moisture, which facilitates oil extraction [64–66] and reduces costs at the olive oil mill. The economic valuation of the investment in irrigation must take into account other benefits, such as avoiding severe droughts in the spring or in the years of installation of the crop. Our results provide information on how to manage the irrigation already installed during the fruit growth phase, a phase in which the published results are highly variable and for which there is no evidence in our agroecological conditions.

In summary, this work provided experimental evidence about the productive behavior of the Arbequina and Frantoio cultivars in a temperate humid climate under different water deficit conditions. We demonstrated that irrigation in a low VPD environment increases fruit weight and the pulp/pit ratio, and that it resulted in a significant increase in yield (kg/tree) in Frantoio. The oil content in response to irrigation was different between cultivars. Water restriction conditions did not affect the oil content of olives in Arbequina, while in Frantoio it increased by water restriction in the evaluated range of stem water potential (from $-0.8$ to $-3.5$ MPa). The content of polyphenols in fruits and in oil increased under water restriction, with lesser changes in other oil quality parameters. A moderate water restriction (50% ETc) produced the most balanced result between yield, oil quality and WP. Irrigation during the growth and ripening of the fruit also affects the vegetative development and therefore will affect the flowering potential for the next season, in this way it is also intervening in the expression of alternate bearing. For this reason, future studies should address aspects of partition and the relationship of vegetative–reproductive growth to carry out a comprehensive analysis of the benefits of irrigation in our agroclimatic conditions.

**Supplementary Materials:** The following supporting information can be downloaded at: https://www.mdpi.com/article/10.3390/agronomy12020313/s1, Figure S1: Annual vapor pressure deficit (mb) in Uruguay and in Spain. Data recorded by an automatic weather station at the experimental site in INIA Las Brujas as an average of the 2009–2020 period (available at http://www.inia.uy/gras/Clima/Banco-datos-agroclimatico 11 January 2022) and by the weather station at Córdoba as an average of the 2001–2020 period (available at https://www.juntadeandalucia.es/agriculturaypesca/ifapa/riaweb/web/estacion/14/6 11 January 2022).

**Author Contributions:** Conceptualization, P.C.-I., C.G., O.B., F.I., R.Z., J.J.V., M.A.-S., I.P.D.L. and G.P.G.-I; formal analysis, P.C.-I. and G.P.G.-I.; methodology, P.C.-I., O.B., F.I. and C.G.; software, C.G. and G.P.G.-I.; investigation, P.C.-I., J.J.V., G.P.G.-I., C.G., M.A.-S. and F.I.; writing—original draft preparation, P.C.-I., G.P.G.-I and M.A.-S.; writing—review and editing, P.C.-I., G.P.G.-I., M.A.-S., O.B., F.I., C.G., J.J.V., I.P.D.L. and R.Z.; supervision, O.B., I.P.D.L. and R.Z.; project administration, P.C.-I.; funding acquisition, P.C.-I., C.G., F.I. and R.Z. All authors have read and agreed to the published version of the manuscript.

**Funding:** This research was funded by The National Institute of Agricultural Research (Instituto Nacional de Investigación Agropecuaria–INIA, Uruguay) (Project INIA FR 22).

**Institutional Review Board Statement:** Not applicable.

**Informed Consent Statement:** Not applicable.

**Data Availability Statement:** Not applicable.

**Acknowledgments:** We are grateful to David Bianchi, Cecilia Martínez, César Burgos and Sergio Bentancor for their collaboration in this project and to Andrés Coniberti for his contribution in the experimental design.

**Conflicts of Interest:** The authors declare no conflict of interest.

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
