# Peer review of "The Impact of Irrigation on Olive Fruit Yield and Oil Quality in a Humid Climate"

_agronomy, doi:10.3390/agronomy12020313_

Round 1

Reviewer 1 Report

Brief summary

This study deals with the impact of irrigation on olive fruit yield and oil accumulation in a non-traditionally cultivated area (Uruguay), under humid climate. In this area the vapor pressure deficit during summer is lower than in traditionally cultivated countries with a Mediterranean climate. Under the current context of climate change, with a growing tendency for extreme events to occur, three irrigation treatments were applied from the end of pit hardening period until harvest.  In two of them an irrigation of 50 and 100% of maximum crop evapotranspiration (ETc) was applied, followed by a third one in which neither irrigation nor rain imputes occurred, in order to evaluate the production response of olive trees to irrigation. From the results a better understanding of the response of olive trees grown under a humid climate to irrigation was achieved.

However, the submitted manuscript needs a major revision. I have listed major concerns below.

Major comments

  1. L. 26: rephrase as follows: “and a third treatment in which neither irrigation nor rain imputes occurred from the end of pit hardening period until harvest”.
  2.  L. 27: rephrase as follows: “from the end of pit hardening period until harvest”.
  3.  L. 54: replace “information that confirm” by “studies which confirm”
  4. L. 55: change “yields” to yield”
  5.  L. 59: change “Fruit drop” to “fruit set”
  6.  L. 65-66: rephrase as follows: “but there is a knowledge gap on the responses of olive trees to irrigation management in humid temperate climates”
  7.  L. 67: change “oil composition” to “oil accumulation”
  8.  L. 72: change “with the” to “in”
  9.  L. 75: change “farms” to farm”
  10.  L. 79-80: the authors need to demonstrate with clarity the treatments of the experiment, i.e. they provided 100% of ETc (control treatment) …
  11.  L. 82: please mention any differences regarding crop load
  12.  L. 103: Did you use the same Kc value throughout the cultivated season? Please specify the value you used?
  13.  L. 106-107: what do the authors mean by “…values used to calculate water balance, according to [25]”? do they refer to the crop coefficient (Kc)? It is not clear. Please rephrase.
  14.  L. 110-111: “that the depletion-water root zone was between field capacity and readily available water”. The authors should provide further information regarding filed capacity and permanent wilting point.
  15.  L. 116-117: this is not the standard procedure for measuring stem water potential. Putting the leaf in plastic bags must precede and in terms of scientific clarity should be mentioned in detail.
  16. Figure 1a: what do the vertical bars indicate? Standard error, standard deviation. Please add this piece of information in the legend below.
  17.  L. 154-160: replace “determined” by “recorded” throughout the paragraph
  18.  L. 173: rephrase as follows: “according to Minguez-Mosquera et al., [33]. Likewise, in Line 176.
  19. L. 176: rephase “of each olive oil”.
  20.  L. 208: replace “Na2CO3” by “Na2CO3
  21.  L. 215: “irrigation treatments are installed after that phase”. Rephrase.
  22.  L. 216-217: “As significant differences were detected” this is a result and should be mentioned there.
  23. Figure 5 and Figure 6: since the authors study the correlation between parameters, R2 should be mentioned in each diagram.
  24.  L. 231: “In the 2021 season, water potential values were more negatives than in the 2019 season”. Please give an (short) explanation regarding the reason why 2021 had more negative values in Discussion.
  25.  L. 315-318: In this paragraph a reduction of 27.3 and 11.8 g GAE/kg FW in total phenols in fruit was recorded, while in the graph a reduction of 2730 and 1180 g GAE/kg FW in total phenols in fruit was observed in Arbequina. The same problem was observed in Frantoio. Please check.
  26.  L. 364-367: this section should be mentioned in Materials and Methods and not on Discussion.
  27.  L. 374-375: rephrase as follows: “These climatic differences can influence olive tree physiology affecting productive variables.”
  28.  L. 399-402: rephrase as follows: “Moreover, a positive linear relationship between the pulp/pit ratio and water plant status was observed in both cultivars (Figure 5 b). This is in accordance with previous studies by Gómez-Rico et al. [45] and Lavee et al. [44] who observed that the irrigation increases the pulp/pit ratio in comparison to rain-fed trees.”
  29.  L. 417-421: rephrase as follows: “In this study a different response in oil content according to cultivars was observed, in agreement with Iniesta et al. [39]. In particular, oil content in Frantoio was higher in the non-irrigated treatment than in the 100 % ETc treatment (Table 2), while in other works a reduction of oil contents has been recorded when stem water potential was near - 4 MPa [45,48].”
  30.  L. 430: “Polyphenols have been associated with defense mechanisms against water stress”. Please add a reference.

Author Response

Dear editors and reviewers,

Thank you very much for your detailed corrections to improve our article and for your quick response, below we detail the responses to each of the comments received.

Reviewer 1.

  1. Done
  2. Done
  3. Done
  4. Done
  5. Done
  6. Done
  7. We write oil composition instead of oil quality to refer to polyphenols, fatty acid profile, pigments, we could change it to oil quality if it is better understood, same in the title.
  8. Done
  9. Done
  10. We rephrase: Three irrigation treatments were applied according to the value of maximum crop evapotranspiration. A first treatment applying 100% ETc, corresponding to full irrigated, a second treatment applying 50% ETc and the third treatment in which neither irrigation nor rain inputs occurred from the end of pit hardening period until harvest. We corrected this in the abstract (L 24) and M&M (L89).
  11. We added (L 98): “The assays were specifically made in years of high fruit load”. It was not carried out in 2020 because it was a year with low fruit load.
  12. Kc values used during the experiment were 0.65 at the beginning of the season and 0.70 during the mid-season and end-season stages. We clarified in M&M (L 129).
  13. Yes, refers to the crop coefficient. We rephrase as (L 131): “kc values used to calculate water balance, according to [25]”.
  14. We added in M&M (L 103): “The soil water curve retention was characterized by measuring water content at ten-sions of 0.01 and 1.5 MPa (field capacity and permanent wilting point respectively), using the Richards and Weaver methods (25). Undisturbed soil sample were used to soil water extraction from different depths up to 0.50 m”.
  15. We added (L 141): “Two hours before the measurement, the shoot was covered with a plastic bag which was then closed, allowing the leaf water potential to balance with the stem water potential a more stable value than that of the leaf water potential.”
  16. Figure 1a: vertical bars indicate standard deviation. We corrected in the legend.
  17. Done
  18. Done
  19. We change for: “the olive oil sample”
  20. Done
  21. We change for: “Irrigation treatments are installed after fruit set.”
  22. We add in results (L 307): Significant differences between the treatments of fruit number per tree at harvest were detected, so productive variables were analyzed with ANCOVA, with fruit number per tree as a covariate.
  23. We appreciate the observation, we incorporate the R2 values in each panel.
  24. We added in results (L 265): “During spring and summer of 2021 there were few rain events. The estimated evapo-transpiration during those months was higher than the precipitation. The crop water demand exceeded water supply from the rains”.
  25. We corrected: a reduction of 2730 and 1180 mg GAE/kg FW in total phenols in fruit was recorded in Arbequina in 2019 and 2021, respectively. Same in Frantoio (470 and 610). We corrected de units in the figure 6 as mg not g.
  26. Done
  27. Done
  28. Done
  29. Done
  30. We add the follow reference: Šamec et al., 2021 “The role of polyphenols in abiotic stress response: The influence of molecular structure”.

Reviewer 2 Report

General comments

I am worried about the over-irrigation result. Adding 410 mm of irrigation water at the 3rd olive fruit phase is huge and costly and non-sense normally.

The abstract is somehow misleading. The positive effect of large volume irrigation had not a significant positive economic effect.

Also, the 4 trees-replicates per treatment cause a defect to the whole research design and results.

All over the manuscript, you try to use the correct units. Thus, you must stick the % after the number it follows throughout the manuscript. This is the correct scientific writing.

Specific comments

L27 add: … end of pit hardening …

L 29 (and L490-1) Actually, no irrigation increased oil content and you must write it this way. In two olive cultivars for olive oil, increasing olive pulp without much oil inside is loss of money due to the cost of irrigation water and its application. So, in the abstract it must be clear that irrigation in this environment is not worthy!!!

L78 what are the four replicates? Trees, blocks of ? trees? Later on it is clear for 4 trees per treatment (L 254).

L88 unclear: 3 sensors in different depths or 2 sensors and the 1st somewhere between 15 to 30 cm?

L132 Well, the total deficit spring and summer is 250 mm, which I accept as logical. Then, after pit hardening you applied 410 mm of irrigation water, which seems extremely high (2-3 fold higher than expected) for your conditions. What did you try to do? To fill soil with excess water, did you cause flood? In addition, this irrigation regime costs lots of money for electricity and has a large environmental cost.

L134-141 but you covered the soil with plastic film at February or January (I guess in January based on 120 d measurements …)? So plants had lots of available water in Febr 2020 or were negatively affected from drought earlier on during fruitlet development (in both springs)?

Fig. 1 Finally, water deficit higher than precipitation was present only in Feb 19 and Jan 21. All other months, precipitation was high or enough for olive trees. And, despite this, you added a tremendous amount of irrigation water.

L163 … ground …

L167 kg of fruit (the ha is not required)

L222 what is CLWP? Describe here, earlier than L240

Fig. 3 2021 season, you have 2 lines with circles and only one with triangles. Which is the 50% irrigation? The same in Fig. 4.

L156 and Fig. 4 You describe you calculated fruit dry matter and here you present only moisture content. Change accordingly.

Fig. 6 It is impossible to have thousands of g of phenols per kg of fresh fruit weight!

L332 clarify the sentence

Table 3 many differences occurred between the two years especially in oleic and linoleic acids, but also in carotenoids, phenolics. Do you explain these in Discussion?

L448 But they found similar results to yours

L463-4 in which treatment?

L467-8 you only measured free fatty acids, not the rest

L478-9 but you talk about fruit phenolics and must present oil phenolics

L494 but shoot growth occurs mainly in spring and flower bud differentiation occurs only in early summer, something way out of your experimental planning.

L571, 573-4 the capitals in title words are not required. The same in some other references.

Author Response

Dear editors and reviewers,

Thank you very much for your detailed corrections to improve our article and for your quick response, below we detail the responses to each of the comments received.

Reviewer 2.

General comments

Comment 1) We apologize for not having written more clearly. Prior to the installation of the experiment, the crop was irrigated according to the value of maximum crop evapotranspiration, so that once the treatments were started, the soil was at field capacity. The plastic was placed to prevent the entry of rain to the soil, and it was placed at the end of December. We did not remove it until harvest, so that the rain from January to May was not available for the plants and therefore did not affect the treatments. The 410 mm was the only water supply during the period to cover 100% of the ET0. We clarified in M&M (L 111).

Comment 2) We agree that only the benefits of irrigation in this phase surely do not justify the investment in irrigation. The economic valuation of the investment in irrigation must take into account other benefits such as avoiding severe droughts in the spring or in the years of installation of the crop. Our results provide information on how to manage the irrigation already installed during the fruit growth phase, a phase in which the published results are highly variable and there no evidence in our agroecological conditions. We clarified in discussion (L 540).

Comment 3) We understand that it would be better to use a greater number of repetitions. However, with that “n” the models were significant and the coefficients of variation were within adequate values, less than 20.

Comment 4) Done. We stick the % after the number it follows throughout the manuscript.

Specific comments

Comment 5) L 27. Done.

Comment 6) L 32 and L 549. We changed for in the abstract and in the discussion: “The response in oil content to irrigation was different within cultivars. Water restriction conditions did not affect the oil content of olives in Arbequina, while in Frantoio it increased it.”

We added in the abstract (L 36): “The technological applicability of the results obtained must be accompanied by an economic analysis.”

Comment 7) L 78. We apologize for not having written more clearly. It is a random complete design. There are no blocks. The experimental unit is the tree and there are 4 trees per treatment, for each cultivar. We added this sentence in M&M (L88).

Comment 8) L 107. We clarify it in the manuscript. There are 3 sensors, each one at a different depth, one at 15 cm, another at 30 cm and another at 45 cm.

Comment 9) L 132. The replacement of irrigation water was carried out according to the balance of water in the soil, it must also be taken into account that there is no contribution of rainwater. Excess irrigation that caused runoff was never recorded, in addition the FDR sensors indicated that they remained within the ranges of water available for that soil.

Comment 10) L 111. Prior to the installation of the experiment, the crop was irrigated according to the value of maximum crop evapotranspiration, so that once the treatments were started, the soil was at field capacity.

Comment 11) L163. Done.

Comment 12) L167. Done. kg of fruit, we removed ha.

Comment 13) We explained CLWP in L 145: The measured SWP values were accumulated over the irrigation period and the cumulative leaf water potential (CLWP) was calculated to compare the level of water stress throughout the entire experiment [29].

Comment 14) I ask if you can give us a few more days to correct figures 3 and 4 because I do not have access to the program these days due to the time of the year. I hope this is not a problem.

Comment 15) L 156 and 162.Done. 

Comment 16) We corrected the units as mg/kg.

Comment 17) L 332. Done.

Comment 18) We added in discussion (L 493): “Despite the fact the levels found in both seasons are different, they are in concordance with previously reported oils from similar experiments. In humid climate conditions the differences between two growing seasons could affect not only the oil content but also minor oil component such as phenolic compounds and fatty acid profile”.

Comment 19) L448. Yes. We clarified.

Comment 20). Done. We added (L 517): “The MUFA/PUFA ratio decreased concomitantly for Arbequina for 2019 season in non-irrigated treatment”.

Comment 21) L467-8. Done. We removed the sentence

Comment 22) L478-9. We corrected.

Comment 23) L494. We refers not only to the number of shoots, but also to the number of new nodes that develop during the summer, which is one of the factors that affects alternate bearing.

Comment 24) L571, 573-4. Done.

Round 2

Reviewer 1 Report

Dear authors,

After your latest revise, the submitted manuscript is significantly improved and can be published.